# Real-World Treatment Outcomes of Different Sequencing Options with Daratumumab, Lenalidomide, and Dexamethasone in Patients with Transplant-Ineligible Multiple Myeloma in Japan

**DOI:** 10.3390/cancers17091389

**Published:** 2025-04-22

**Authors:** Kazuhito Suzuki, Yuma Fujimori, Chika Sakai, Hiroaki Tsuchiya, Yosuke Koroki

**Affiliations:** 1Division of Clinical Oncology and Hematology, Department of Internal Medicine, The Jikei University School of Medicine, Tokyo 105-8461, Japan; kaz-suzuki@jikei.ac.jp; 2Johnson & Johnson, Tokyo 101-0065, Japan; csakai1@its.jnj.com (C.S.); htsuchi7@its.jnj.com (H.T.); ykoroki@its.jnj.com (Y.K.)

**Keywords:** daratumumab, lenalidomide, newly diagnosed multiple myeloma, overall survival, real-world evidence, treatment sequence

## Abstract

Daratumumab in combination with lenalidomide + dexamethasone (DRd) is used for transplant-ineligible newly diagnosed multiple myeloma (NDMM). However, there is limited evidence on when DRd should be introduced in the treatment sequence. Using Japanese claims data, we conducted a retrospective study to identify the optimal treatment strategy for DRd administered as first-line (1L), second-line (2L), and third-line or later (≥3L) therapy. Among 344 patients with NDMM who received DRd (208 received 1L, 78 received 2L, and 58 received ≥3L), overall survival (OS) and time to next treatment (TTNT) were analyzed. First-line DRd demonstrated a trend toward improved OS from the time of MM diagnosis (inverse HR of 0.658; 95% CI: 0.278, 1.563) and prolonged TTNT from the first DRd administration (inverse HR of 0.746; 95% CI: 0.481, 1.149) compared with 2L DRd. These results suggest that the use of 1L DRd may provide the best outcomes for transplant-ineligible NDMM patients.

## 1. Introduction

Multiple myeloma (MM) is a hematologic malignancy characterized by monoclonal proliferation of plasma cells that leads to bone disease, including lytic lesions and osteoporosis [1,2]. In Japan, MM accounts for about 10% of hematopoietic malignancies and causes approximately 4000 deaths each year [2].

Although treatment options have increased [3,4,5,6], and novel drugs have been shown to improve overall survival (OS) in Japanese myeloma patients [7], optimal management of patients with MM remains challenging, particularly regarding treatment selection and sequencing. A real-world US-based study showed that transplant-ineligible newly diagnosed MM (NDMM) patients had increasing attrition rates in later treatment lines, highlighting the importance of selecting the optimal regimen as early as possible [8,9,10,11].

Daratumumab is a human IgG1κ mAb targeting CD38. Multiple regimens containing daratumumab are standard of care in Japan [2]. Daratumumab + lenalidomide + dexamethasone (DRd) and daratumumab + bortezomib + dexamethasone (DVd) were approved in Japan in September 2017 for patients with relapsed or refractory MM (RRMM) [12]. In August 2019, daratumumab + bortezomib + melphalan + prednisolone (DVMP) was approved in Japan for transplant-ineligible NDMM, and in December 2019, DRd was also approved for this indication [12]. Subsequently, approval was granted in Japan for daratumumab + carfilzomib + dexamethasone (DKd; November 2020) [12], and daratumumab + pomalidomide + dexamethasone (DPd; December 2021) for RRMM. A subcutaneous formulation of daratumumab is also available in Japan (approved in March 2021) [13].

DRd is the only regimen available in Japan for the treatment of both NDMM and RRMM, and it is increasingly used in 1L in Japanese clinical practice [14]. Its efficacy in MM as first-line (1L) and second-line (2L) or later treatment was demonstrated in the MAIA [15] and POLLUX studies [16]. In the interim analysis of the phase 3 MAIA study in patients with transplant-ineligible NDMM, DRd was associated with significantly improved OS (hazard ratio [HR] 0.68; *p* = 0.0013) and progression-free survival (PFS; HR 0.53; *p* < 0.0001) compared with lenalidomide + dexamethasone (Rd) alone after a median follow-up of 56.2 months, and significantly prolonged PFS on next line of treatment (PFS2) compared with Rd alone (HR 0.61; *p* < 0.0001) [15]. The recently released long-term follow-up analysis of the MAIA study confirmed these findings, with a 33% reduction in the risk of death with DRd versus Rd (HR 0.67; *p* < 0.0001) at a median follow-up of 89.3 months [17]. In the phase 3 POLLUX study, patients with RRMM who had received ≥1 prior lines of therapy were treated with DRd or Rd [16,18,19,20]. At a median follow-up of 44.3 months, DRd significantly prolonged PFS compared with Rd (HR 0.44; *p* < 0.0001) [20]. Furthermore, a significant improvement in OS was observed with DRd versus Rd at 79.7 months (HR 0.73; *p* = 0.0044) [19]. The survival benefits in both these trials were maintained in subgroups defined by older age (≥75 years in the MAIA study [15] and ≥65 years in the POLLUX study [19]).

In a simulation study of data from the MAIA study and real-world US data from the Flatiron Health database, patients with transplant-ineligible NDMM had a prolonged median OS with 1L DRd compared with 2L daratumumab administration after bortezomib + Rd (VRd) or Rd [21]. This suggests that 1L DRd may provide greater benefits over daratumumab-based regimens in later lines of treatment in this population.

A database study in Japan found a shift in MM treatment patterns in recent years, with shorter 1L treatment durations and a move toward regimens that incorporate newer drugs in subsequent lines [3]. Despite the POLLUX study demonstrating prolonged PFS and OS with 2L DRd compared with Rd [19,20], it remains controversial whether DRd should be used as 1L or 2L treatment to achieve optimal clinical outcomes in transplant-ineligible NDMM, and there is very limited real-world evidence regarding the treatment sequencing of DRd in MM.

This retrospective database study was conducted to identify the optimal timing for DRd in patients with transplant-ineligible NDMM (1L or later lines) in real-world settings in Japan. The key objectives were to investigate the impact of the different DRd treatment lines on OS and time to next treatment (TTNT) in these patients.

## 2. Materials and Methods

### 2.1. Study Design and Data Source

This was a retrospective, multicenter, non-interventional study of patients who were newly diagnosed with MM between 1 January 2016 and 31 October 2023 and who received DRd treatment (Figure A1). The study used retrospective, anonymized claim data from the Medical Data Vision (MDV) database (Medical Data Vision Co., Ltd., Tokyo, Japan). The MDV database comprises standardized healthcare, hospital-based insurance claim data in Japan, using the Japanese Diagnosis and Procedure Combination fixed-payment reimbursement system, covering approximately 30,000 patients with MM [3]. The database has been used previously to study treatment patterns and clinical outcomes in Japanese patients with MM [22,23].

The study design included a baseline period of ≥12 months before MM diagnosis (with the date of confirmed MM diagnosis considered the index date) and a follow-up period from the index date to death or the end of available data, whichever occurred first (Figure A2a–c).

The study was conducted in accordance with relevant Japanese legal and regulatory requirements, including those related to scientific purpose, value, and rigor, and followed generally accepted research practices described in the Guidelines for Good Pharmacoepidemiology Practices (GPP). As the study used anonymized data that were retrospectively collected from the MDV database (which itself complies with all relevant Japanese and international requirements concerning patient consent and data use), informed consent from patients to use their data and ethics approval for the study were not required.

### 2.2. Patient Population

Patients who met the following criteria were included in the study: males and females aged ≥20 years with a confirmed diagnosis of MM (according to the International Classification of Disease, tenth version [ICD-10], diagnostic code C900) and a confirmed prescription of DRd for 1L, 2L, or third-line or later (≥3L) treatment of MM, who were considered to have transplant-ineligible MM and had not received auto/allogeneic stem cell transplantation. The patients needed to have had continuous care in the baseline and follow-up periods, and ≥60 days of follow-up data from the index date and ≥12 months of baseline data prior to the index date.

Exclusion criteria included a history of MM treatment before the first MM diagnosis; a history of other primary cancer(s) or metastatic disease in the baseline period; initiation of 1L treatment >365 days after the index date; and receipt of a daratumumab-based regimen before the start of DRd (Figure A1).

Patients were stratified into 1L, 2L, and ≥3L groups based on their DRd line of treatment, as per the treatment patterns defined in a previous MDV database study of treatments and outcomes in MM [3]. In addition, the following criteria were applied: discontinuation of a single agent from a combination regimen was not considered to be a change of treatment line; a time gap of >365 days without MM treatment was considered to be the start of the subsequent treatment line, even if the two consecutive regimens were the same; and ixazomib monotherapy was considered as maintenance therapy if commenced immediately after 1L treatment (and after March 2020, its approval date in Japan) and analyzed in combination with 1L therapy.

### 2.3. Outcomes

The main study outcomes included OS from the time of MM diagnosis in each treatment-line group (Figure A2a); TTNT from the first DRd administration in each treatment-line group (Figure A2b); OS from the first DRd administration in each treatment-line group (Figure A2c); baseline characteristics of patients at MM diagnosis and at the first DRd administration in each treatment-line group; and patterns of treatment regimen usage.

### 2.4. Statistical Analysis

The sample size was based on the total number of patients in the MDV database with NDMM who received daratumumab between January 2016 and June 2023.

Descriptive statistics were used to present patient demographic and baseline characteristics, including number and proportion of patients, mean, standard deviation (SD), median, minimum, and maximum. Missing data were not included.

The probability of event (OS and TTNT) was estimated using the Kaplan–Meier method. OS was defined as the time from MM diagnosis or the first DRd administration to death from any cause, while TTNT was defined as the time from the first DRd administration to the day before the initiation of the next line of treatment or to death (whichever occurred first). For OS, if patients’ survival outcome was unknown, their data were censored at the date that they were last known to be alive. Because information regarding death was only available for hospitalized patients, OS was calculated using inpatient deaths only. For TTNT, data were censored at the last recorded activity in the medical record. If the last record was prescription of DRd, the longest drug grace period in the regimen from the time of the last prescription of the DRd regimen was used for censoring.

HRs and 95% confidence intervals (CIs) in each treatment-line group were estimated using the Cox regression model. Treatment line (1L vs. 2L), sex (male vs. female), age (<75 vs. ≥75 years), Charlson Comorbidity Index (CCI, using the Quan adaption; low vs. medium/high/very high) [24,25], and the presence/absence of comorbidities (congestive heart failure, mild liver disease, diabetes without chronic complications, diabetes with chronic complications, and renal disease) were added as covariates in the multivariate analyses. The selection of comorbid diseases for the multivariate analysis was based on the relationship between comorbidites and MM, and the number of patients who had each condition.

Treatment patterns were evaluated using Sankey diagrams [26], with treatments categorized by mechanism of action (i.e., daratumumab-based, mAb-based, other CD38 antibody-based, proteasome inhibitor [PI] without immunomodulatory drug [IMiD], IMiD without PI, PI with IMiD, and others).

Data extraction from the MDV database was performed using Amazon Redshift (Amazon.com, Settle, WA, USA) and SQL. All other analyses were conducted using SAS version 9.4 (SAS Institute, Cary, NC, USA) and R version 4.3.1 (R Core Team, R Foundation for Statistical Computing, Vienna, Austria).

## 3. Results

### 3.1. Study Population

In total, there were 35,581 patients in the MDV database with a new diagnosis of MM between 1 January 2016 and 31 October 2023, of whom 682 started 1L therapy within 365 days of MM diagnosis. Baseline and demographic data were available at the time of MM diagnosis for 344 patients (Figure A3). Of these patients, 208 (60.5%) received DRd as 1L, 78 (22.7%) as 2L, and 58 (16.9%) as ≥3L (Table 1). The median (range) age was 76 (51–92) years in the overall population and was similar across treatment-line groups. The median time from MM diagnosis to DRd administration was 47 days overall, and was 20 (range 0–354), 217 (16–1520), and 452 (64–2211) days in patients who received DRd as 1L, 2L, and ≥3L, respectively. The CCI score was categorized as ‘low’ (score of 0) or ‘medium’ (score of 1 or 2) in 79 (38.0%) and 86 (41.3%) patients, respectively, in the 1L group, 24 (30.8%) and 37 (47.4%) patients in the 2L group, and 21 (36.2%) and 22 (37.9%) patients in the ≥3L group.

In total, baseline and demographic data were available at the time of the first DRd administration for 323 patients (Table 2). Of these, 200 patients (61.9%) received 1L DRd, 72 (22.3%) received 2L DRd, and 51 (15.8%) received ≥3L DRd. The baseline and demographic characteristics of these patients were consistent with those of the overall population.

### 3.2. Overall Survival

The median OS from the time of MM diagnosis was not reached (NR) in the 1L and 2L groups (median follow-up of 430.5 and 813.0 days, respectively) and was 70.2 months (95% CI: 48.8, NR) in the ≥3L group (median follow-up of 996.0 days; Figure 1). At 36 months, the OS rate was 0.862% in the 1L group, 0.792% in the 2L group, and 0.736% in the ≥3L group. A trend toward longer OS was observed in the 1L versus 2L group (as indicated by the reciprocal (inverse) HR of 0.658; 95% CI: 0.278, 1.563). When assessed from the first DRd administration, median OS was NR in the 1L and 2L groups (median follow-up of 390.0 and 551.0 days, respectively) and was 36.6 (95% CI: 23.5, NR) months in the ≥3L group (median follow-up 448.0 days; Figure A4).

### 3.3. Time to Next Treatment

The median (95% CI) TTNT from the first DRd administration was 36.5 months (26.2, NR) in the 1L group, 21.6 months (16.3, NR) in the 2L group, and 13.9 months (8.2, 20.6) in the ≥3L group (Figure 2). There was a trend toward prolonged TTNT in the 1L versus 2L group (inverse HR of 0.746; 95% CI: 0.481, 1.149).

### 3.4. Univariate and Multivariate Analyses

According to the analysis of potentially prognostic factors for OS, the use of 1L DRd appeared to confer survival advantages. There was a numerically higher risk of worse survival from the time of MM diagnosis with 2L treatment versus 1L treatment in both the univariate (HR 1.52; 95% CI: 0.64, 3.60) and multivariate (HR 1.56; 95% CI: 0.65, 3.75) analyses (Table A1). Similarly, there was a numerically higher risk of worse survival with ≥3L versus 1L DRd treatment in the univariate analysis (HR 1.66; 95% CI: 0.71, 3.88).

In the univariate analysis of potentially prognostic factors for TTNT from the first DRd administration, there was a trend toward shorter median TTNT in patients receiving 2L versus 1L DRd treatment (HR 1.34; 95% CI: 0.87, 2.08) and in patients receiving ≥3L versus 1L DRd treatment (HR 2.25; 95% CI: 1.44, 3.50; Table A2). Similar results were observed in the multivariate analysis (2L vs. 1L treatment, HR 1.34; 95% CI: 0.85, 2.12; Table A2).

### 3.5. Treatment Patterns

The Sankey diagram of treatment patterns of daratumumab-based and other regimens is shown in Figure 3.

Among 208 patients in the 1L DRd group, 205 patients (98.6%) received 1L DRd treatment alone and three patients (1.4%) received 1L DRd followed by ixazomib maintenance; 51 (24.5%) went on to receive other 2L treatment, and 17 (8.2%) subsequently received other ≥3L treatment. The most common 2L treatments in this group were daratumumab-free treatments (received by 66.7% of patients, most commonly Rd [15.7%], Vd [13.7%], and Pd [7.8%]). However, 33.3% of patients received daratumumab-based treatments as 2L, including DVd (13.7%), other daratumumab-based treatments (15.7%), and DPd (3.9%). Among the 17 patients who subsequently received 3L treatment, a slight majority received daratumumab-based treatments (52.9%; including four patients [23.5%] who received DRd), while the remaining received daratumumab-free (47.1%) treatments.

Of the 78 patients in the 2L DRd group, all received 2L DRd treatment, and 67 (85.9%) subsequently received other ≥3L treatment. Among the patients in the ≥3L DRd group (*n* = 166), 58 (34.9%) received ≥3L DRd. In the 2L DRd group, the most common other 1L regimens (in ≥20% of patients) were PI without IMiD (Vd 33.3%), PI with IMiD (VRd 28.2%), and IMiD without PI (Rd 25.6%). In the ≥3L DRd group, the most common other 1L regimens were PI without IMiD (Vd 43.1%) and IMiD without PI (Rd 22.4%), and the most common other 2L regimens were IMiD without PI (Rd 27.6%) and PI with IMiD (VRd 20.7%).

## 4. Discussion

In this real-world study of patients with transplant-ineligible NDMM in Japan, those who received 1L DRd showed a trend toward an improvement in OS from the time of MM diagnosis by approximately 34% compared with 2L DRd (inverse HR of 0.658; 95% CI: 0.278, 1.563). Furthermore, by using 1L DRd, rather than delaying its administration until 2L treatment, a trend toward prolongation of TTNT from the first DRd administration by approximately 25% was observed (inverse HR of 0.746; 95% CI: 0.481, 1.149). These findings were confirmed in analyses of potential prognostic factors for OS and TTNT, in which a numerically higher risk for worse OS and shorter median TTNT with 2L and ≥3L versus 1L DRd treatment was observed. Overall, these results suggest that DRd is more effective when used in earlier treatment lines.

These findings are consistent with the improved PFS outcomes observed in clinical studies of DRd when used as 1L in the MAIA study [15] or 2L in the POLLUX study [20]. POLLUX evaluated the efficacy of DRd used as 2L or later and showed that PFS was prolonged with fewer lines of therapy [20], an outcome that, along with our findings, may support the hypothesis that early treatment with a regimen that provides longer TTNT translates into better OS outcomes for patients with MM. In cases where CD38 mAb therapy is not utilized until 2L, its application in later lines remains valuable. POLLUX demonstrated superior PFS with DRd regardless of prior lines of therapy [20].

In addition, the Cox regression analysis in our study demonstrated that the effect of treatment line (1L vs. 2L) on outcomes was consistent even after adjusting for factors associated with OS and TTNT.

The findings of our Japanese real-world study are consistent with those from the previous simulation study of clinical and real-world US data from patients with transplant-ineligible NDMM and real-world Singapore data [21,27]. In the simulation study, 1L DRd (followed by 2L pomalidomide- or carfilzomib-based regimens) prolonged the median OS by 1.0–2.7 years compared with 1L VRd (followed by a 2L daratumumab-based regimen) and by 2.3–3.8 years compared with the 1L Rd (followed by a 2L daratumumab-based regimen) [21]. Although there are concerns that the results of such simulation studies may not be directly applicable to clinical practice, in part because attrition rates may not be reflective of those in clinical practice, the trends observed in our study do substantiate the simulation study findings, in that 1L DRd was associated with survival benefits. This was despite differences in study methodology and patient populations (i.e., in our study, the 2L and ≥3L groups only included patients able to receive later lines of therapy who were of a younger age).

In the MAIA study, 17.9% of the study participants in the DRd arm (1L treatment) were aged ≥80 years [28], while in the POLLUX study, 10.1% of the patients in the DRd arm (2L or later treatment) were aged ≥75 years [16]. In the current study, patients receiving 1L DRd were older (median age of 76 years) than those receiving 2L or ≥3L DRd (75 and 73 years, respectively). This indicates that, in clinical practice, DRd is used in patients older than those participating in clinical studies [29]. Despite the older age of patients in our study, a trend toward improved OS and TTNT with 1L versus 2L DRd was demonstrated, suggesting that there is a benefit of early treatment with DRd in clinical practice irrespective of patient age. This is consistent with the findings from the POLLUX and MAIA studies, which also found survival benefits in older age groups (≥65 years [19] and ≥75 years [15]). Furthermore, a sub-analysis of the MAIA study conducted to assess the effects of treatment in groups of patients defined by a frailty status (i.e., fit, intermediate, or frail) found that DRd was effective regardless of the frailty status [28]. The MAIA study also demonstrated the improvement in health-related quality of life over a 5-year period in DRd compared with Rd in transplant-ineligible NDMM patients [30].

This study does have some limitations that need to be considered when interpreting the results. As this study used retrospective data from the hospital claim-based MDV database, information about death could only be obtained if the patient died during hospitalization; as a result, it is possible that OS was underestimated. In addition, patients who transferred to another hospital were considered lost to follow-up. However, such patient transfer would have been at random and is, therefore, unlikely to have affected the reliability of the HRs obtained in our study. Laboratory data type and content were limited, with treatment response assessments not recorded. There are no direct data in the database to classify patients as transplant-eligible or transplant-ineligible. Therefore, we used the treatment record of auto/allogeneic stem cell transplantation, along with inclusion and exclusion criteria, to identify patients likely to be transplant-ineligible. The study recruitment period was between January 2016 and October 2023; however, DRd was approved for 1L treatment of transplant-ineligible NDMM in December 2019 (i.e., approximately half-way through the study), a fact that is likely to have influenced the number of patients who received DRd as 1L treatment in the study cohort. In addition, the short follow-up period may have limited the ability to adequately detect differences in OS and TTNT, particularly for first-line DRd. Longer follow-up periods could strengthen the conclusions drawn from the study. Finally, in Japanese clinical practice, treatment may be changed before disease progression; however, this could not be determined in our study and may have influenced the reported outcomes, particularly TTNT, leading to a divergence of results compared with those reported in the MAIA study [15]. There is a possibility that patients who transitioned to 2L treatment due to inadequate response or intolerance to DRd in the 1L setting were included, potentially underestimating TTNT outcomes in the 1L group. Among patients receiving DRd as 2L therapy, there may have been cases where treatment was administered not as salvage therapy but due to insufficient therapeutic effect or intolerance in the 1L setting. This may have led to an overestimation of OS and TTNT outcomes in the 2L group.

## 5. Conclusions

In this real-world study of patients with transplant-ineligible NDMM in Japan, OS from the time of MM diagnosis and TTNT from the first DRd administration showed a trend toward improvement when DRd was used as 1L therapy versus 2L or later treatment. These findings support those of a previous treatment sequence modeling study using real-world data from Japan and reinforce the potential of using DRd in 1L to maximize patient benefit.

## Figures and Tables

**Figure 1 cancers-17-01389-f001:**
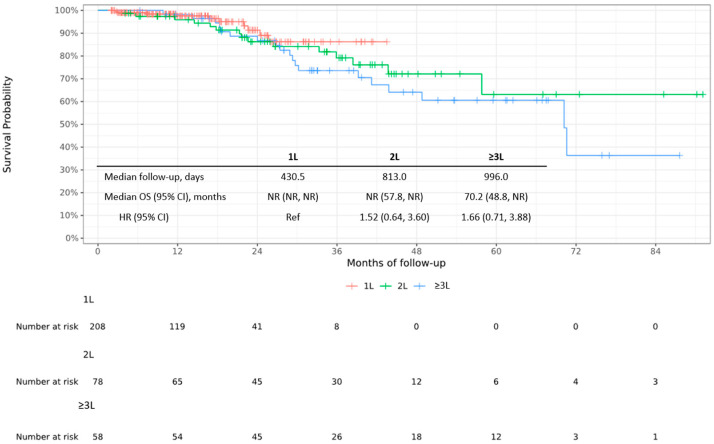
Kaplan–Meier curve of overall survival stratified by DRd treatment line from time of diagnosis of multiple myeloma. 1L, first line; 2L, second line; ≥3L, third line or later; CI, confidence interval; DRd, daratumumab + lenalidomide + dexamethasone; HR, hazard ratio; NR, not reached; OS, overall survival; Ref, reference.

**Figure 2 cancers-17-01389-f002:**
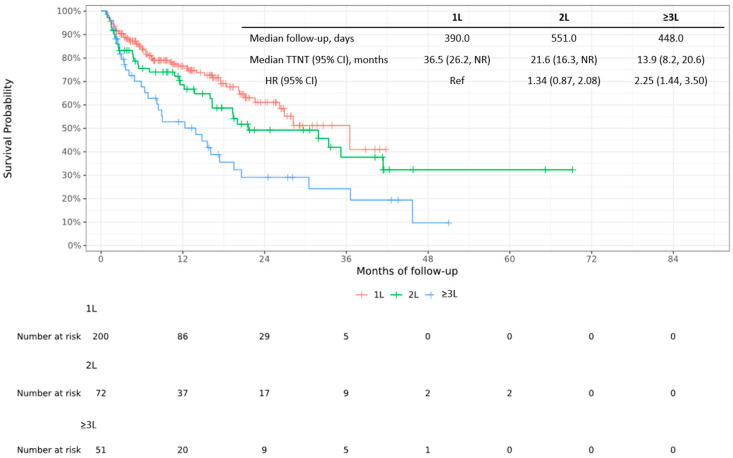
Kaplan–Meier curve of time to next treatment stratified by DRd treatment line from the first DRd administration. 1L, first line; 2L, second line; ≥3L, third line or later; CI, confidence interval; DRd, daratumumab + lenalidomide + dexamethasone; HR, hazard ratio; NR, not reached; TTNT, time to next treatment; Ref, reference.

**Figure 3 cancers-17-01389-f003:**
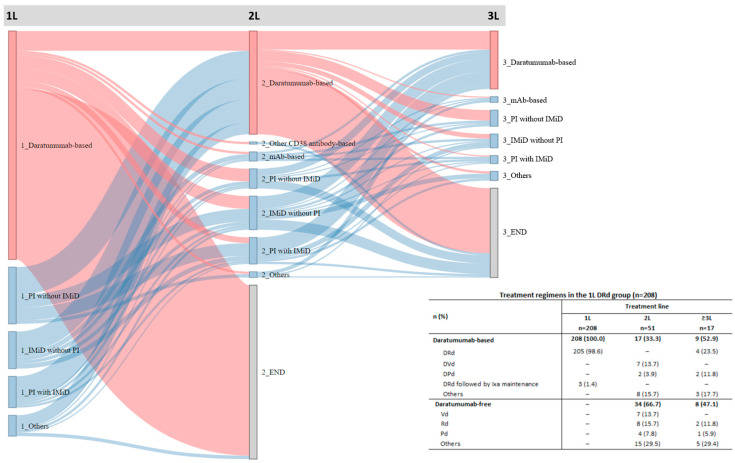
Sankey diagram of treatment patterns stratified by daratumumab treatment line. 1L, first line; 2L, second line; 3L, third line; DPd, daratumumab + pomalidomide + dexamethasone; DRd, daratumumab + lenalidomide + dexamethasone; DVd, daratumumab + bortezomib + dexamethasone; IMiD, immunomodulatory drug; Ixa, ixazomib; mAb, monoclonal antibody; Pd, pomalidomide + dexamethasone; PI, proteasome inhibitor; Rd, lenalidomide + dexamethasone; Vd, bortezomib + dexamethasone.

**Table 1 cancers-17-01389-t001:** Patient baseline characteristics at the time of multiple myeloma diagnosis, stratified by daratumumab + lenalidomide + dexamethasone treatment line (*N* = 344).

Parameter	1L DRd *n* = 208	2L DRd *n* = 78	≥3L DRd *n* = 58
Sex, *n* (%)			
Male	103 (49.5)	45 (57.7)	33 (56.9)
Female	105 (50.5)	33 (42.3)	25 (43.1)
Age, median (range), years	76 (51–90)	75 (52–92)	73 (51–92)
CCI category, *n* (%)			
Low (score 0)	79 (38.0)	24 (30.8)	21 (36.2)
Medium (score 1 or 2)	86 (41.3)	37 (47.4)	22 (37.9)
High (score 3 or 4)	35 (16.8)	15 (19.2)	15 (25.9)
Very high (score ≥ 5)	8 (3.8)	2 (2.6)	0
Time from diagnosis to DRd initiation, median (range), days	20 (0–354)	217 (16–1520)	452 (64–2211)
BMI, mean ± SD, kg/m^2^	22.8 ± 4.2	23.5 ± 3.4	23.2 ± 3.1
Comorbidities, *n* (%)			
Congestive heart failure	40 (19.2)	17 (21.8)	13 (22.4)
Renal disease	38 (18.3)	13 (16.7)	7 (12.1)
Diabetes	31 (14.9)	13 (16.7)	8 (13.8)
Without chronic complications	25 (12.0)	10 (12.8)	6 (10.3)
With chronic complications	6 (2.9)	3 (3.8)	2 (3.4)
Cerebrovascular disease	29 (13.9)	6 (7.7)	10 (17.2)
Liver disease	25 (12.5)	10 (12.8)	6 (10.3)
Mild	24 (11.5)	10 (12.8)	6 (10.3)
Moderate or severe	1 (0.5)	0	0
Chronic pulmonary disease	20 (9.6)	9 (11.5)	2 (3.4)
Peptic ulcer disease	18 (8.7)	16 (20.5)	15 (25.9)
Peripheral vascular disease	12 (5.8)	5 (6.4)	4 (6.9)
Rheumatic disease	7 (3.4)	5 (6.4)	2 (3.4)
Myocardial infarction	5 (2.4)	2 (2.6)	2 (3.4)
Dementia	2 (1.0)	2 (2.6)	1 (1.7)
Hemiplegia or paraplegia	2 (1.0)	0	0
AIDS/HIV infection	1 (0.5)	0	0

1L, first line; 2L, second line; ≥3L, third line or later; AIDS, autoimmune deficiency syndrome; BMI, body mass index; CCI, Charlson Comorbidity Index; DRd, daratumumab + lenalidomide + dexamethasone; HIV, human immunodeficiency virus; SD, standard deviation.

**Table 2 cancers-17-01389-t002:** Patient baseline characteristics at the first daratumumab + lenalidomide + dexamethasone administration, stratified by line of treatment (*N* = 323).

Parameter	1L DRd *n* = 200	2L DRd *n* = 72	≥3L DRd *n* = 51
Sex, *n* (%)			
Male	99 (49.5)	41 (56.9)	30 (58.8)
Female	101 (50.5)	31 (43.1)	21 (41.2)
Age, median (range), years	77 (51–91)	76 (52–92)	74 (52–94)
CCI category, *n* (%)			
Low (score of 0)	54 (27.0)	2 (2.8)	4 (7.8)
Medium (score of 1 or 2)	92 (46.0)	36 (50.0)	18 (35.3)
High (score of 3 or 4)	37 (18.5)	23 (31.9)	21 (41.2)
Very high (score of ≥5)	17 (8.5)	11 (15.3)	8 (15.7)
Time from diagnosis to DRd initiation, median (range), days	16 (0–354)	214 (28–1520)	420 (77–1863)
BMI, mean ± SD, kg/m^2^	22.47 ± 3.30	22.39 ± 3.03	22.36 ± 3.36
Comorbidities, *n* (%)			
Congestive heart failure	51 (25.5)	26 (36.1)	21 (41.2)
Renal disease	41 (20.5)	24 (33.3)	12 (23.5)
Diabetes	45 (22.5)	21 (29.2)	15 (29.4)
Without chronic complications	37 (18.5)	16 (22.2)	11 (21.6)
With chronic complications	8 (4.0)	5 (6.9)	4 (7.8)
Cerebrovascular disease	37 (18.5)	17 (23.6)	12 (23.5)
Liver disease	30 (15.0)	14 (19.4)	10 (19.6)
Mild	29 (14.5)	14 (19.4)	10 (19.6)
Moderate or severe	1 (0.5)	0	0
Chronic pulmonary disease	35 (17.5)	26 (36.1)	16 (31.4)
Peptic ulcer disease	30 (15.0)	26 (36.1)	27 (52.9)
Peripheral vascular disease	12 (6.0)	8 (11.1)	2 (3.9)
Rheumatic disease	7 (3.5)	5 (6.9)	3 (5.9)
Myocardial infarction	7 (3.5)	6 (8.3)	3 (5.9)
Dementia	2 (1.0)	1 (1.4)	2 (3.9)
Hemiplegia or paraplegia	2 (1.0)	0	0
AIDS/HIV infection	1 (0.5)	0	0

1L, first line; 2L, second line; ≥3L, third line or later; AIDS, autoimmune deficiency syndrome; BMI, body mass index; CCI, Charlson Comorbidity Index; DRd, daratumumab + lenalidomide + dexamethasone; HIV, human immunodeficiency virus; SD, standard deviation. Patients who met the exclusion criteria between the time of their multiple myeloma diagnosis and the first administration of DRd were excluded.

## Data Availability

The data sharing policy of Johnson & Johnson is available at https://www.janssen.com/clinical-trials/transparency (accessed on 1 April 2025). These data were made available by Medical Data Vision Co., Ltd., and used under license for the current study and are not publicly available. Other researchers should contact https://en.mdv.co.jp/.

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
