# Peer review of "Real-World Treatment Outcomes of Different Sequencing Options with Daratumumab, Lenalidomide, and Dexamethasone in Patients with Transplant-Ineligible Multiple Myeloma in Japan"

_cancers, 2025, doi:10.3390/cancers17091389_

Round 1
Reviewer 1 Report
Comments and Suggestions for Authors
This is a pertinent treatment paper on treatment of myeloma. It is a thorough retrospective analysis that may influence the choice of initial treatment. Some questions include- why were patients transplant ineligible? Minor issues are: Figures need a reference check; Section 3.4 needs to be placed in line with other sections; clarify placement of data in text vs appendices; a list of abbreviations and definitions would be helpful,
Comments on the Quality of English LanguageThe paper has useful information for clinicians and researchers. Several issues for clarification are listed in author section above. The research question and methods/analysis are appropriate. Conclusions are reasonable for a retrospective study. The paper does need some revisions for clarification.
Author Response
Comments 1: Why were patients transplant ineligible?
Response 1: Thank you for raising this question. We would like to address this point in the limitation section of Discussion part. We acknowledge the limitation of this study to determine whether each patient is clearly transplant-eligible or transplant-ineligible directly from the database. To address this, we applied several factors and criteria to include patients who were considered likely to be transplant-ineligible:
- We included patients who had not received auto/allogeneic stem cell transplantation
- The patients needed to have had continuous care in the baseline and follow-up periods, and ≥60 days of follow-up data from the index date and ≥12 months of baseline data from before the index date
Patients with the age ≥65 years old and/or severe comorbidities are typically considered as transplant-ineligible, and based on the screened patients’ baseline characteristics, most of the patients included in this study are likely representative of transplant-ineligible patients.
We have added this point to the discussion section in the manuscript as an additional limitation:
“There are no direct data in the database to classify patients as transplant-eligible or transplant-ineligible. Therefore, we used the treatment record of auto/allogeneic stem cell transplantation along with specific inclusion and exclusion criteria to identify patients likely to be transplant-ineligible.”
Comments 2: Figures need a reference check
Response 2: Thank you for pointing this out. We re-confirmed all the figures and captions to eliminate any errors.
Comments 3: Section 3.4 needs to be placed in line with other sections
Response 3: I agree with the suggestion. Section 3.4 is switched with the section 3.5 to make the structure to enhance readability.
Comments 4: clarify placement of data in text vs appendices
Response 4: We appreciate this suggestion. After reviewing the manuscript, we found that the descriptions provided in the main text as well as in the figures’ and tables’ captions clearly indicate where the data are presented. We believe that the manuscript clarifies the data placement.
Comments 5: a list of abbreviations and definitions would be helpful
Response 5: Thank you for this comment. In our manuscript, all abbreviations are defined within the main text, figure legends, and table captions, following the common practice. We did not include a separate abbreviations list to avoid redundancy.

Reviewer 2 Report
Comments and Suggestions for Authors
I havent suggestions for autors. Correctly covered topic.
Author Response
Comments 1: I havent suggestions for autors. Correctly covered topic.
Response 1: Thank you for your feedback. I am delighted to hear that you found the topic to be correctly covered. I appreciate your encouragement and will continue to work on further developing this research.
Reviewer 3 Report
Comments and Suggestions for Authors
Refer attachment.

Author Response
Major Comments 1: The short follow-up time may underestimate differences in OS and TTNT, particularly for first-line DRd. The authors should acknowledge this limitation and discuss how longer follow-up could strengthen the conclusions.
Response to Major Comments 1: Thank you for highlighting this limitation. We agree that the relatively short follow-up period may have influenced the ability to detect differences in OS and TTNT, especially for first-line DRd. To address this, we included the following statement in the limitations section of the Discussion part: “In addition, the short follow-up period may have limited the ability to adequately detect differences in OS and TTNT, particularly for first-line DRd. Longer follow-up periods could strengthen the conclusions drawn from the study.”
Major Comments 2: Changes in treatment due to intolerance (rather than progression) could affect TTNT interpretation. The authors should discuss whether this was accounted for and how it may influence results.
Response to Major Comments 2: Thank you for highlighting this point. We acknowledge that changes in treatment due to intolerance, rather than disease progression, could influence the interpretation of TTNT. This is an important limitation of our study, and we would like to address it in the limitation section under the Discussion part.
Specifically, we included the following statement: “There is a possibility that patients who transitioned to 2L treatment due to inadequate response or intolerance to DRd in the 1L setting were included, potentially underestimating TTNT outcomes in the 1L group. Among patients receiving DRd as 2L therapy, there may have been cases where treatment was administered not as salvage therapy but due to insufficient therapeutic effect or intolerance in the 1L setting. This may have led to an overestimation of OS and TTNT out-comes in the 2L group.”
Major Comments 3: DRd was approved for first-line use midway through the study, potentially biasing the firstline cohort toward later patients. The authors should assess pre- or post-approval trends.
Response to the Major Comments 3: Thank you for raising this important point. There are no clear trends at pre- and post-approval with the following reasons. We would like to clarify that all patients in the 1L DRd cohort were diagnosed with NDMM after the approval of DRd for first-line treatment. Regarding the 2L cohort, approximately one-third of the patients were diagnosed prior to the approval of 1L DRd. It is important to note that Rd for TIE NDMM was approved in December 2015. Setting January 2016 as the starting point for the analysis allows for the inclusion of patients who could have been treated with Rd and Vd, which are comparable to the current treatment options for 1L TIE NDMM, aside from regimens containing daratumumab. Furthermore, daratumumab with lenalidomide became available in September 2017 for RRMM. Consequently, patients newly diagnosed with MM after January 2016 would be eligible to enter the study as 2L DRd-treated patients. We hope this explanation addresses the concern and provides clarity on the timeline and rationale underlying the study period.
Minor comments 1: Table 1 (diagnosis) and Table 2 (first DRd administration) report slightly different patient numbers (344 vs 323). A footnote explaining exclusions (e.g., missing data) would prevent confusion.
Response to minor comments 1: Thank you for pointing out this discrepancy. We have added the following footnote to Table 2, to clarify the difference in patient numbers between Table 1 (diagnosis) and Table 2 (first DRd administration): "* Patients who met the exclusion criteria between the time of their multiple myeloma diagnosis and the first administration of DRd were excluded.”
Minor comments 2: The manuscript states that transferred patients were censored but does not quantify how many were lost. Reporting attrition numbers would improve transparency.
Response to minor comments 2: Thank you for your valuable comment. For enhancing transparency, we have ensured that censored patients, including those lost-to-follow-up, are represented as tick marks in the Kaplan-Meier curves, indicating their inclusion in the analysis. The comment is considered as one of the limitations of the database, we are unable to quantify the exact number of patients who were transferred to other hospitals. The MDV database does not capture information regarding hospital transfers, and as a result, such patients are considered lost-to-follow-up within the dataset. Similarly, patients who passed away at home are also categorized as lost-to-follow-up, making it challenging to estimate the number of transferred patients specifically, so we would like to hand censored data with the marks in the figures.
Minor comments 3: Figures are blurry and, in some cases, text is not readable.
Response to minor comments 3: We have reviewed and updated all the figures to ensure they are clear and visually accessible. The text within the figures has also been adjusted to improve readability.
